# Research Progress on Rolling Forming of Tungsten Alloy

**DOI:** 10.3390/ma17184531

**Published:** 2024-09-14

**Authors:** Jun Cao, Jie Xia, Xiaoyu Shen, Kexing Song, Yanjun Zhou, Chengqiang Cui

**Affiliations:** 1School of Mechanical and Power Engineering, Henan Polytechnic University, Jiaozuo 454000, China; xjovetyy@163.com; 2Zhejiang Tony Electronic Co., Ltd., Huzhou 313000, China; jerry.shen@tonytech.com; 3Henan Academy of Sciences, Zhengzhou 450046, China; kxsong@haust.edu.cn; 4School of Materials Science and Engineering, Henan University of Science and Technology, Luoyang 471000, China; dazhou456@163.com; 5State Key Laboratory of Precision Electronic Manufacturing Technology and Equipment, Guangdong University of Technology, Guangzhou 510006, China; cqcui@gdut.edu.cn

**Keywords:** tungsten alloy, process route, thermomechanical processing, rolling process, micro-structure, texture evolution

## Abstract

Tungsten is a metal with many unique characteristics, such as a high melting point, high hardness, high chemical stability, etc. It is widely used in high-end manufacturing, new energy, the defense industry, and other fields. However, tungsten also has room-temperature brittleness, recrystallization brittleness, and other shortcomings due to the adjustment of the composition and organizational structure, such as the addition of alloying elements, adjusting the phase ratio, the use of heat treatment and deformation strengthening, etc. Its performance can be improved to meet the requirements for use in different fields. At present, the main production method of tungsten alloy is powder metallurgy. The use of a rolling open billet rotary forging–stretching process can improve production efficiency and product quality, but in actual production, due to the combined effects of various factors, such as elastic deformation of rolling elements, plastic deformation of the rolled material, etc., the mechanical properties of tungsten plates and bars are often difficult to control effectively, seriously affecting rolling stability and production efficiency. For this reason, researchers have conducted extensive and deep research and optimization on the rolling process of tungsten alloys, including establishing mathematical models, performing numerical simulations, optimizing process parameters, etc., providing important references for the rolling and forming of tungsten alloys. Meanwhile, the material properties are greatly influenced by the microstructure, and the evolution of the microstructure can be well quantified by some advanced characterization techniques, such as SEM, TEM, EBSD, etc., so that certain properties of tungsten can be obtained by controlling the texture evolution. In conclusion, this paper comprehensively summarizes the research progress of tungsten alloy roll forming and provides an important reference for further improving the processing performance and production efficiency of tungsten alloy.

## 1. Introduction

Tungsten (W) is a rare, high-melting-point metal known for its excellent physical and chemical properties. With high density (19.3 g/cm), high melting point (3422 °C), high strength (3 GPa), high hardness, high thermal conductivity, strong abrasion resistance, strong corrosion resistance, etc., it has an irreplaceable role in the fields of the national defense industry, the nuclear industry, and the civil industry [1,2,3]. Although tungsten is considered as a potential candidate material for divertor and plasma-facing material (PFM) components in future fusion reactors due to its excellent physical properties, such as low hydrogen solubility and high sputtering threshold energy [4,5,6,7,8,9], the low-temperature brittleness (ductile–brittle transition temperature is higher than 300 °C), irradiation brittleness, recrystallization brittleness (recrystallization brittleness occurs at 1200 °C), low heat-load tolerance, and irradiation-induced hydrogen retention of pure W constitute limitations of the candidate materials, and technological innovations and technological breakthroughs are urgently needed to unleash the full potential of W [10].

The ductile–brittle transition temperature (DBTT) of pure W is generally high, often exceeding 300 °C [11], so W alone is brittle at room temperature, and pure W can create large grid cracks or even fracture completely during testing [12]. Research shows that the fracture behavior of W depends not only on the composition of the alloy, but also on the manufacturing details of the material (such as strain rate and surface finish) and microstructural characteristics (such as impurities within the grains, dislocation density, grain morphology, and texture) [13,14,15]. Brittleness is more common in body-centered cubic (BCC) crystal structures, which is mainly attributed to the relatively few independent slip systems within this structure [16]. First, the main influence on the low-temperature brittleness of W is the inner lack of dense rows of planes. As a typical BCC metal, the plastic deformation mechanism of W significantly depends on the migratory ability of nonplanar 1/2<111> screw dislocations [17], and the complex diffusion mechanism of 1/2<111> screw dislocations makes it difficult for the three-dimensional structure to move smoothly along the established slip systems in the BCC structure [18]. Thus, the BCC structure, due to its limited number of slip systems and the unideal balanced relationship between yield strength and fracture strength, leads to significant brittle behavior at low temperatures. Secondly, another key factor for the low ductility of W alloys lies in its weak grain boundary (GB) properties, as they are closely related to the unavoidable concentration of solutes such as O, C, P, and N [19,20]. These impurities are mainly concentrated in the grain boundary (GB) region [21], where the high degree of crystal disorder at the grain boundaries and the segregation of the impurity elements (O and N) are particularly significant, which together weaken the strength of the grain boundaries and thus become the key factors leading to brittle damage and deterioration of the strength properties of the materials [22]. Therefore, altering the impurity distribution and enhancing the grain boundaries will be effective in improving strength and ductility. In this case, the most widely used route is to refine its microstructure [23]; this method not only significantly enhances the final strength of the material, but also optimizes the uniformity of the impurity distribution by generating dense grain boundaries, which in turn enhances the ductility of the material.

Based on the Peierls–Nabarro stress theory and the modified Hall–Petch and Cottrell–Petch relationships, the strategy of increasing dislocation width and grain refinement can effectively enhance the slip capacity and improve the DBTT [24]. Grain refinement techniques mainly include solid solution strengthening and second-phase dispersion [25,26]. These methods cover the solute drag effect, Orowan stress, and pinning effect, which effectively retard the migration of grains and grain boundaries, and thus enhance the material properties. Over the past decades, three core strategies have emerged to improve the mechanical properties of W, especially ductility and fracture toughness: alloying [27,28], synthesis of composites [29,30], and plastic deformation strengthening techniques [31,32,33]. The most common W alloy additions are aimed at improving ductility, which is achieved by increasing dislocation mobility [17], removing interstitial impurities [34], refining the microstructure, or increasing the recrystallization temperature [35,36,37,38,39]. First, in the field of alloying, Re alloying is an effective strategy to enhance the ductility of tungsten materials. Research shows that the formation of W-Re solid solution using the solution-softening method can significantly reduce the Peierls stress in W and activate more slip planes, thereby optimizing the dislocation core structure, accelerating the migration of spiral dislocations at low temperatures, and significantly improving the plastic performance of the material [40,41]. In the meantime, more economical elements such as Ta [42], V [43], and Ti [44] replace Re in order to achieve a similar strengthening effect. In addition, the use of second-phase strengthening strategies is an effective way to enhance the properties of W alloys from the material design perspective [45]. In recent studies, some process strategies have focused on the homogeneous dispersion of oxide nanoparticles into the W matrix to prepare high-performance nanostructured W materials [46,47]. The dispersed oxide particles effectively inhibit grain boundary slip and hinder grain growth, thus enhancing the strength, toughness, and recrystallization properties of tungsten alloys [48,49]. For rare-earth oxides, since rare-earth resources are expensive and scarce, some ceramic oxide materials and carbides (such as Al_2_O_3_, ZrO_2_, ZrB_2_, TiC, and TaC) have been created as substitutes for rare-earth oxides [50,51,52,53], and carbides, with their higher melting points and excellent thermal stability, are ideal for enhancing the erosion resistance of W alloys [54,55]. Second, for the synthesis of composites, foils and fibers are the only form of composites among polycrystalline W materials that exhibit reliable room temperature ductility under additive-free conditions. In order to extend this desirable mechanical behavior of foils to bulk shapes, Reith et al. [56,57] proposed and investigated the fabrication of W laminate composites by assembling multiple layers of ductile W foils. The W laminate, consisting of 20 layers of 0.1 mm thick rolled tungsten foils joined by eutectic silver–copper brazing compounds, exhibited excellent low-temperature toughness with a DBTT of less than 300 °C, which is significantly lower than that of most reported pure W materials. Although previous work has shown that dispersed particles such as TiC, ZrC, and Y_2_O_3_ can improve the strength of tungsten materials, the conventional dispersion-reinforced tungsten sintered bodies still present a large number of disadvantages such as intergranular fractures, low density, low strength, and poor plasticity in the absence of hot-pressing processing. Finally, the limited ductility of the material is mainly attributed to the weak grain boundary cohesion, high residual porosity, insufficient dislocations, and laminated grain structure of the sintered state material. It is found that the plastic deformation strengthening technique can effectively regulate the number and distribution of defects and alloy structure in W alloys and thus significantly improve the comprehensive performance of the materials. Miao et al. [34] found that the DBTT of tungsten without thermo-mechanical treatment, only by SPS sintering and addition of 0.2 wt% TaC, was reduced from 700 °C to 650 °C. After further increasing TaC to 0.5 wt% and supplementation with 70% rolling, the DBTT was significantly reduced to about 250 °C. Therefore, thermomechanical processing has been found to be the most effective way to increase W plasticity and reduce BDTT [58]. Previous studies have shown that a variety of thermomechanical processing techniques, such as rolling [59,60,61], die forging [62], rotary forging [63], high-pressure torsion [64], and high-pressure torsion [65,66], can enhance the ductility of tungsten materials to varying degrees.

Most studies on thermo-mechanically processed tungsten have shown that optimization of microstructure texture, strengthening of grain boundaries, formation of layered structures, and high dislocation densities with improved dislocation mobility together constitute the core advantages of rolling to enhance tungsten ductility. At present, the tungsten billet rolling open billet method is roughly divided into two-roll rolling, planetary rolling, four-roll rolling, and a three-roll Y-type continuous rolling mill. Among them, tungsten open billet is most widely used in actual production by using a three-roll Y-type mill to roll tungsten open billets continuously in multiple passes, and the reasonable design of a three-roll Y-type mill’s hole type and continuous rolling process can effectively avoid the defects produced by the traditional rotary forging processing method [67]. In actual production, the high tensile strength and high hardness of 800–1000 MPa of powder-sintered tungsten billets, the temperature of thermomechanical processing, the plastic deformation of rolled materials, the vibration of the rolling mill, the rolling process parameters, and other factors make it difficult to effectively control the grain size, surface quality, and product dimensional accuracy of the alloy. Therefore, the mechanical properties of tungsten plates, rods, and wires are reduced. Due to the above reasons, the current domestic production process of tungsten alloy makes it difficult to produce high-performance products, which seriously affects the rolling stability and production efficiency [68]. This paper reviews the research progress of tungsten alloy roll forming, which provides a reference basis for further optimizing the processing technology and use performance of tungsten alloy.

## 2. Pre-Rolling Process Manufacturing Route

Traditionally, powder metallurgy has remained the predominant method of producing tungsten and its alloys due to the relatively low rate of material formation of melted alloys, the coarse grain size of billets, and other defects. The improvement in the performance of tungsten and its alloys makes it difficult to form high-performance products solely by the rolling process in continuous processing technology. Its pre-manufacturing process route is also particularly important. In pre-rolling, it still has to go through a series of processes involving micro-powder manufacturing technology, densification technology, and continuous processing technology.

### 2.1. Micro-Powder Manufacturing Technology

The micro-fine powder manufacturing process is the key to tungsten material performance enhancement. Micron or nano-scale powders directly determine the internal organization and structure distribution of the material from the microscopic level. The key aspects of tungsten materials which have excellent properties are grain refinement, organization homogenization, and second-phase particle homogenization, which are directly dependent on the micro-powder manufacturing technology. To achieve a fine-grained microstructure, ultrafine and uniform nano-powder is the first choice. For ductility, ultrafine grain (UFG) or nanostructured W materials can significantly enhance the ductility of W and effectively reduce its ductile–brittle transition temperature (DBTT) [69]. For toughness, the addition of alloying elements or second-phase particles is an efficient strategy to enhance the toughness of W materials [70]. At present, tungsten-based high-density alloy powder preparation methods include a spray-drying method, a mechanical mixing method (mechanical alloying), a condensation-drying method, chemical methods (sol–gel method, solution combustion synthesis method, and wet chemical method), a gas-phase precipitation method, a reaction jet method, a vacuum plasma jet deposition method, a mechanical–thermal chemical synthesis method, etc. Mechanical alloying and chemical methods are still the most used for micro-powder manufacturing in most studies. The mechanical mixing method is used to form pre-alloyed mixed powders via collision of metal balls on the powder, which triggers plastic deformation, cold welding, and crack generation and refines the crystal blocks to the nanometer scale. Deformation reshapes the particle morphology, cold welding induces particle consolidation and enlargement, and cracking refines the particles, which together contribute to the homogeneous refinement of alloys and composites [71]. Mechanical ball milling is usually used to prepare tungsten alloy powders doped with alloying elements such as Re [72], Ta [73], and V [74]. Although the mechanical alloying technique is capable of producing nanocomposite powders with particle sizes of less than 20 nm, highly homogeneous atomic-level mixing, and hard materials (such as carbide) for the ball milling jars and balls, the process is still prone to the introduction of impurities, such as C and Co, and the powders are prone to aggregation and adherence to the walls of the equipment, which may adversely affect the quality of the final product [75]. For this, tungsten balls of the same material are used as ball milling media, which can be used to prepare higher-purity powder, and a small amount of process control agents such as alcohol, carbon tetrachloride, and stearic acid can be added to the ball milling process to effectively inhibit the phenomenon of sticking to the wall and aggregation. Among the chemical methods, Zheng Chen et al. [76] introduced solution combustion synthesis as a novel method for the preparation of tungsten nano-powders. The results show that the inhibition of particle densification is enhanced with the increase in the addition of La_2_O_3_ to the second-phase particles. The minimum grain size of the sample with 2.0 wt% La_2_O_3_ added is 0.47 μm, and the maximum microhardness value is 739.3 Hv0.2, which achieves a good effect. Alloys prepared by powder metallurgy (PM) techniques are considered as diffusion-strengthened alloys [77], and diffusion strengthening belongs to the category of second-phase strengthening, which depends on the chemical and thermodynamic stability of the second phase, desirable morphology, extremely small size, sufficient quantity, and good diffusivity. In diffusion-strengthened alloys, dislocations need to move around large particles through a bypass mechanism, and these large particles are stabilized and embedded by advanced techniques such as mechanical alloying, which can be as small as about 1 μm in size. Orovan’s theory states that dislocations encounter strong particle-generated resistance as they move, leading to a cumulative increase in stress during successive deformations to overcome the resulting reverse stress. According to the dislocation theory, the higher the number of particles and the smaller the spacing between particles, the more pronounced the strengthening effect. Currently, oxides and carbides are the main components of the reinforced phases, where oxides are mostly rare earth oxides (La_2_O_3_ [78], Y_2_O_3_ [79], and HfO_2_ [80]), metal–ceramic oxides (Al_2_O_3_ [81] and ZrO_2_ [82]), and carbides (TaC [83], ZrC [84], and TiC [85]). Oxide-reinforced materials are favored due to high-temperature stability and low price [86]; for example, in rare-earth lanthanum- and yttrium oxide-doped tungsten, ultrafine La_2_O_3_ particles are uniformly dispersed, which significantly refines the W microstructure and enhances the mechanical properties of W alloys. Carbides, on the other hand, are important in composites for a wide range of applications due to their high hardness, high melting point, and excellent thermal stability. Currently, the methods of doping rare earth oxide particles into refractory metals are solid–solid, liquid–solid, and liquid–liquid, respectively. For example, Zhang Jun et al. [87] prepared W-Lu_2_O_3_ alloys by doping Lu_2_O_3_ particles into a tungsten matrix through ball milling using the solid–solid method. Yar et al. [88] prepared W-1.0 wt%Y_2_O_3_ and W-0.9 wt%La_2_O_3_ nano-powders by reacting yttrium nitrate or lanthanum nitrate with ammonium sec-tungstate in aqueous solution using the solid–liquid method. Liu et al. [89] produced W-1.0 wt%Y_2_O_3_ powder via the sol–gel method using the liquid–liquid approach. Liquid–liquid doping is processed at the molecular level to generate the most homogeneous nanoscale microstructures, giving the materials ultra-high strength and excellent ductility. The commonly used doping routes are mostly focused on the fabrication of micron-level tungsten alloys, which have lower properties than nanoscale tungsten alloys [90]. Therefore, in recent years, there have been many research avenues for the preparation of nano-powders, such as hydrothermal synthesis [91,92], the co-precipitation method [93], the sol–gel method [89], mechanical alloying [49], azeotropic distillation [94], and solution combustion synthesis [95], which have successfully prepared nanoscale tungsten powders [96]. These studies focus on the use of oxide and carbide additives to enhance the mechanical properties of tungsten through grain refinement and inhibition of recrystallization. Under the given conditions, the high mobility of powder particles facilitates sintering densification [97], nano-sized powder particles exhibit higher sintering activity compared to micron-sized ones [98], and powders with a normal distribution are easier to densify than those with a uniform particle size [99].

### 2.2. Densification Technology

Due to the micro-powder manufacturing process, the powder particles are not completely combined, and there are certain holes and pores which affect the organization and properties of the material to a certain extent, so to obtain the fine-grain organization and high-density characteristics of the W materials, densification technology is also needed to achieve this. In the densification stage, the sintering temperature and holding time are the most important factors in obtaining a high-density block, while poor control can also cause damage to the material. Currently, W densification techniques cover a wide range of methods, such as spark plasma sintering (SPS), ultra-high pressure (UHP), hot isostatic pressure (HIP), hot press (HP), and pressure-less sintering (PLS). Because pressure sintering has the presence of pressure, it is easier to obtain fine-grain and high-density W blocks than pressure-less sintering. SPS is one of the most efficient techniques for achieving low-temperature densification of refractory metals rapidly [100]. SPS effectively inhibits the growth of powder particles and promotes the formation of fine-grained materials with its high heating rate (50–1000 °C/min), additional pressure, and current [101]. Zhou Z. et al. [102] used the SPS technique to solidify the W-0.5 wt%Y_2_O_3_ composites at 1700 °C, achieving a relative density of 96.8% in only 1 min, with the grain size controlled between 3 and 5 μm, during which an auxiliary pressure of 50 MPa was applied. The most important difference between the technical characteristics of UHP and SPS is that UHP allows for higher auxiliary applied loads [103]. Zhang X et al. [104] successfully solidified pure W powders with particle sizes of 0.3 μm and 3 μm, respectively, using ultrahigh-pressure technology under 7 GPa pressure. The relative densities of the pure W materials could reach 95.1% and 98.1%, respectively, under the condition of a 2800 A current lasting for 60 s. After UHP densification, the grain size of W remains essentially unchanged, with the original 0.3 μm grain size slightly increasing to 0.33 ± 0.1 μm, whereas for the initial 3 μm pure tungsten powders, the grains are significantly refined to 1.85 ± 0.84 μm and 0.47 ± 0.2 μm, with the latter exhibiting a bimodal distribution. This phenomenon is attributed to the fact that the UHP technique promotes intergranular diffusion and plastic flow while effectively suppressing atomic diffusion, thus hindering excessive grain growth. HIP technology is a special sintering method in which a raw or encapsulated raw billet is placed in a vacuum vessel to achieve a densification process through the synergistic action of the high temperature and all-around high pressure [105]. Kurishita et al. [106] utilized a HIP device to solidify ball-milled W-TiC composite powders for 3 h at temperatures ranging from 1620 K to 2200 K and at a pressure of 200 MPa. The results showed that the relative density of W-TiC composites was as high as 98%, and the grain size was refined to 0.9 μm. In contrast to the technologies above, HP technology relies mainly on uniaxial pressure assistance for high-temperature densification. Therefore, the mechanical properties of W materials densified only by HP technology need to be further improved by subsequent plastic deformation processes such as rolling. Densification of W materials is often carried out using the PLS method, which requires high temperature and a longer processing time, but this method is prone to lead to coarse grains with poor mechanical properties. Especially for large-sized samples, it is more difficult to achieve the preparation of W materials with fine-grain microstructure [107]. Ren et al. [108] succeeded in refining the grain size of pure W materials to the submicron level by solidifying nanoscale powders in the range of 1000–1300 °C for 1 h under Ar or H_2_ atmospheres using the PLS technique. However, limited by the weight (only 4 g) and size (~16 mm in diameter) of the sample, it is not easy to prepare W materials with nanoparticle fine-grain organization on a large scale. It is difficult to achieve complete densification and standardized properties of W materials directly, either by pressure sintering or pressure-less sintering. Therefore, the densification of W materials is often combined with subsequent processing techniques to further enhance their properties through the construction of fine crystallization and high-density tissues.

### 2.3. Continuous Processing Technology

By adjusting the type and quantity of W and added elements, or by adding compounds to tungsten alloys, adjusting the ratio of W and bonded phases, and obtaining tungsten alloys with different properties with the help of heat treatment and deformation strengthening techniques, tungsten alloys with different properties can be used to satisfy the requirements of multiple fields of use [109]. Although heat treatment can improve the properties of tungsten alloys, obtaining ultra-high-strength tungsten alloys needs to be realized with the help of deformation strengthening, etc. [110]. The subsequent deformation processing technology is not only the key way to refine the grain and enhance the density of tungsten materials, but also an effective strategy to improve the comprehensive performance of tungsten materials. The four kinds of grain refinement, grain orientation change, grain morphology change, and dislocation density increase are the main influences on deformation processing [41]. Grain refinement brings high-density grain boundaries and more deformation-optimized grains, which significantly improves the strength and toughness of W materials [111,112]. Grain orientation and morphology confer anisotropy to material properties. A moderate increase in dislocation density, especially movable screw-type dislocations, enhances the dislocation activity and, thus, the plasticity and toughness of the material [113]. Zhao [114] and Rupp et al. [115] found that, in a three-point bending test of rolled W material, longitudinal specimens were fractured by through-crystal disintegration, while transverse ones showed intergranular fracture. The fracture toughness in the longitudinal direction was better than that in the transverse direction, and this phenomenon was attributed to the dual effects of textural changes and crystal morphology. Currently, in order to enhance the properties of tungsten materials (covering pure tungsten; tungsten–tantalum alloys; tungsten–rhenium alloys; and composites of tungsten with ZrC, Al_2_O_3_, La_2_O_3_, and others), a range of deformation machining techniques, such as forging, rolling, rotary forging, and extrusion, are widely used in the industry. These techniques, through the implementation of specific process steps and procedures, aim to optimize the organization of the material and thus significantly enhance its overall properties. This process not only deepens the adjustment of the material’s microstructure, but also promotes the overall improvement of its macro properties.

## 3. Rolling Mill and Pass Design

### 3.1. Two-Roll Rolling Mill and Pass Design

Since the melting point of tungsten is too high, it is not possible to obtain tungsten alloys directly by the casting method, but they can be prepared by powder metallurgy process. The traditional tungsten alloy preparation process involves mixing W powder and alloy powder, molding, densification by the sintering process, and then a series of treatments and post-processing of sintered products. Among them, the fiber organization of tungsten and its alloys is closely related to the mixing state of the original powder; if the original powder is not homogeneous, it is easy to produce segregation and compositional inhomogeneity in the sintered organization, forming holes, which is not conducive to the densification of tungsten alloys [116]. Therefore, alloys prepared from powders with high purity, small grain size, and good homogeneity often have excellent properties. However, during the sintering process, due to grain growth and its own pore space, the sintered material will lead to lower density, lower strength, and poor plasticity, and thus, it cannot meet the requirements for certain heavy-duty use. Liu Mingcheng [117] pointed out that the plastic-processing strengthening process can effectively regulate the number and distribution of defects in tungsten alloys and the alloy structure and then comprehensively enhance the comprehensive performance of the material. At present, pure tungsten and its alloys are mainly realized by rolling, spin forging, drawing and other ways to achieve plastic forming, while hydrostatic extrusion and intense plastic deformation (such as isotropic angular extrusion and high-pressure torsion) technology is also used to strengthen tungsten-based materials.

Due to the limitations of the traditional rotary forging process, in 1960, the Austrian company Poulan succeeded in producing molybdenum wire by rolling in a two-roll mill. This opened a new era of rolling processes to replace the rotary hammer process for the production of tungsten and molybdenum wire, and its production process has been improved and has been used until now [118]. Rolling can be used for the processing and deformation of plates and bars, using the rolling process to form different materials according to the flow characteristics of the material to determine its processing parameters [119]. Two-roll rolling is the main way to obtain tungsten plates. Tungsten plasticity at room temperature is very poor; the temperature is more than 1000 °C before plastic processing, so it is suitable for hot rolling and warm rolling; it is difficult to carry out cold rolling [120]. Tungsten plate series products mainly include hot-rolled plates, cold-rolled plates, strips, and foils, while the two-roll mill is mainly used for hot rolling of tungsten plates and bars. The two-roll mill has square, rhombus, oval, and round pass types, etc. For tungsten plates, square pass types are mainly used for rolling. The whole rolling process is divided into four stages: end biting, tugging, stabilizing rolling, and end of rolling. Each pass of rolling consists of these four processes, and the stabilization rolling process is the most important forming stage, which directly affects the metal flow of the plate, the stress–strain distribution process, and the final quality control. The end nibbling stage is the first condition for the rolling process to be carried out. The subsequent rolling process cannot be carried out if nibbling cannot be realized. The conveyor table is usually used in the actual production to ensure that nibbling is completed smoothly and to improve the efficiency at the same time [121].

Beginning in the 1960s and 1970s, researchers have studied the main process technology of W plates produced by the powder metallurgy method. Xiao Songtao et al. [122] used a thermal simulation test machine to test and establish a mathematical model of the deformation resistance of tungsten plates in the stress state coefficient model, which provides a reference basis for the rolling production process. Shen Hong et al. [123,124] utilized DEFORM-3D v5.0 software to carry out numerical simulation of the tungsten plate-rolling process for stress–strain and damage situations. Combined with the actual production, it was concluded that the comprehensive performance of tungsten plate rolling was better at a heat treatment temperature of 1250 °C and a deformation of 30%. In addition, a Smart Crown roll was designed to study the rolling of tungsten plates by its axial movement, and it was found that the new process was superior to the traditional process. By comparing the hot-rolling process regimes of one fire and one pass, one fire and two passes, and one fire and three passes, Liu Ningping et al. [125] found that adopting one fire and two passes by reducing the time between the two passes of rolling could ensure the temperature of slabs, improve the plasticity of slabs, and lead to finer and more uniform fiber tissues. However, since the process is based on advanced processing equipment and the speed of manual operation, it is not applicable to all tungsten plate-processing processes. Gao Xing et al. [126] found that intense extrusion deformation caused nanoscale pure tungsten to form an ultrafine crystalline structure with high-energy and large-angle non-equilibrium grain boundaries. These features promote the redistribution of impurity elements at the grain boundaries, improve the plasticity of pure tungsten, reduce the tough-brittle transition temperature, and enhance the processability. The production of ultrafine crystalline tungsten has become one of the key ways to reduce the brittleness and improve the deep processing capability. Liu Qiangqiang et al. [127] used software to simulate and analyze the rolling process of tungsten plates of different thicknesses and simulated the adjustment of process parameters to combine a more ideal rolling process. Yang Xiaowei et al. [128] studied the rolling process of large-sized tungsten plates. By comparing and selecting three different rolling processes, the process of rolling was carried out by adopting the process of hot-rolling open billet with one fire for one pass and warm rolling of the finished product with one fire for two passes, and finally, a large-sized tungsten plate of 0.5 mm × 600 mm × 700 mm was successfully produced. Its surface quality was good, the density reached 19.25 g/cm, and the room-temperature bending angle was more than 30°. Yang Zhuoyue [129] studied the deformation and failure characteristics of 93W-Ni-Fe alloy cylindrical specimens under high-strain-rate compressive loading with the aid of scanning electron microscopy and optical microscopy. It was found that the inhomogeneous deformation of the specimen was regional, cracks were first formed in the region with the highest tensile stress at the time of failure, and the cracks expanded to a certain extent and then polymerized through the shear zone. In service environments, brittle fracture defects occur in tungsten plates. For the fracture toughness of tungsten plates in applications, Riedle and Gumbsch et al. [130,131] have extensively studied the fracture toughness of tungsten single crystals and their interrelationships between crystal orientation, crack extension direction, loading rate, and temperature. Daniel Rupp et al. [132] have argued that the fracture behavior of single-crystal tungsten also applies to polycrystalline tungsten. B. Gludovatz et al. [133] found that the fracture toughness of polycrystalline tungsten increases with temperature in the range of −196 to 1000 °C. Especially in the low-temperature region, the fracture toughness is significantly influenced by the organization and, thus, dependent on the processing. A. Alfonso et al. [134] studied the thermal stability of tungsten plates strongly deformed by hot rolling in the range of 1000 °C to 1250 °C. The results showed that the thickness of hot rolled tungsten plates decreased considerably during this period and that moderate deformation of tungsten plates working at higher temperatures led to excellent stability. Zhang et al. [135] studied the texture evolution and thermo-mechanical properties of pure tungsten under different depressions. It was found that 60% of depressions endowed pure tungsten with excellent thermal workability and Charpy impact properties, while 80% of depressions significantly enhanced its radiation resistance. Q. Wei [136] explored the effect of low-temperature rolling on the tensile mechanical properties of commercially pure tungsten and found that when the rolling temperature was lower than the nominal recrystallization temperature of 1250 °C, the ductility and strength of tungsten could be effectively enhanced. The VALDUC Mechanics Laboratory in France developed the finite element analysis code POLLUX to simulate the upsetting test of tungsten and obtained the intrinsic relationship and friction coefficients in the range of 700~1000 °C, and the simulation results showed good agreement with the experimental results [137].

In modern bar mills, only specific hole designs are used. Horizontal stands are equipped with oval hole types, and vertical stands with round hole types. This layout optimizes the use of live sleeves between the stands for efficient tension-free rolling. In this arrangement, constant hole line and torsion-free rolling can be realized. When the two-roll mill rolls large-size bars, there is a significant difference in the linear velocity of each point within the hole type, resulting in uneven deformation of the surface of the rolled parts, which is prone to cause internal cracking and wear of the roll holes, affecting the quality of the finished product and increasing the consumption of rolls [138]. Based on the challenges of two-roll hot rolling of tungsten plates, such as the narrow temperature range and large deformation resistance, scholars at home and abroad combine experiments and numerical simulations to deeply analyze the key variables such as metal flow, load, stress–strain, strain rate, and temperature distribution in rolling. Thus, the process parameters are optimized to ensure product quality and effectively avoid the cost of test errors.

### 3.2. Three-Roll Rolling Mill and Pass Design

Based on the three-roll Y-type, the rolling mill has the advantages of a high compression rate of passes, continuous processing production, small tolerance of products, and excellent overall performance of the finished products. The use of a three-roll Y-type mill for the production of tungsten and molybdenum rods and wires and its advantages in terms of metal organization and mechanical properties are significantly more than the traditional spin-forging open billet process-produced products. This improves the quality of the processed products and the yield of the subsequent drawing. This means that the Y-type mill is the most widely used in the rolling of hard-to-deform metals. Tungsten bars are rolled in a Y-type continuous rolling unit, which consists of a flexible combination of 4 to 14 stands. Each stand has three built-in round rolls, which are arranged in the space with a precise 120° angle, forming a unique “Y” structure and, thus, the name of a Y-type rolling mill. This design is driven by a DC motor and gearbox, achieving a highly efficient and stable rolling process [139].

In the Y-type mill, its horizontal rolls for the main drive rolls rely on the bevel gear transmission of the other two rolls, with the upper drive and the lower drive alternating their arrangement. Compared with the two-roll mill, the three-roll mill has greater superiority in rolling. Zhang Yijun et al. [140] elaborated on the advantages of three-roll milling as follows. First, the three-roll mill has a small width spread, high deformation rate, low energy consumption, and a low temperature rise. Second, the deformation along the cross-section is uniform, and there is automatic compensation along the length direction. Third, it can achieve “free size” rolling, with the strong hole type utility.

The three-roll Y-type mill originated in the 1950s and was developed by the Italian company CONTINUS-PROPERZI. Its rolling conditions are “single line, no torsion, micro-tension” rolling, also known as “compact mill” and “micro mill”. It is mainly used for continuous casting and rolling of non-ferrous metals such as Cu, Al, and its alloy bars [141]. In the mid-1970s, the West German KOCKS company first developed a three-roll high-speed wire rod mill that could roll tungsten rods and successfully carried out production. The mill’s roll diameter was 250~640 mm, and the rolling speed was 0.1~0.6 m/s for the cross-sectional shape of the round or hexagonal hot-rolling bar. The number of rolling passes could be 3 to 10 passes, the total deformation was about 37% to 94%, and the lengthwise elongation of the rolled parts was about 1.5 times or 16.6 times the original. The roll was made of a high-quality cobalt-based alloy. The roller hole type for a group of three rollers constituted a hexagonal cross-section; each frame of the three rollers was easy to replace, with three rollers for an active roller and two slave rollers. The active rollers drove the slave rollers so that the three rollers synchronized their movement. The roll hole size did not produce excessive errors due to multiple rolling. The development of this tungsten rod three-roll high-speed rolling mill solved the problem of difficult processing of large crystalline blanks. The rolled tungsten rod diameter uniformity was good, the residual stress distribution was more uniform, and the deformation speed was fast. The rolled products had a high yield rate while saving energy and labor resources [142,143]. In the 1980s, the Soviet Union successfully developed the MK-380 four-roll mill and successfully rolled round tungsten–molybdenum rods with billet diameters of 32~65 mm into fine rods with diameters of 16~32 mm. Rolling billets are powder-sintered tungsten billets, which are heated to 1500 ± 50 °C in a hydrogen resistance furnace and then fed into a rolling mill for rolling. The rolling hole types of this mill are square–square hole type, round–square hole type, and square–round hole type [144]. Domestically, the Y-type three-roll mill has also been widely used. In 1970, Luoyang Nonferrous Metals Processing Design and Research Institute, Anshan Iron and Steel Miniature Plant, and Dalian Industrial and Mining Vehicle Plant jointly cooperated to develop China’s first Φ350 mm three-roll Y-type continuous rolling mill [145]. It has rolled steel, copper, aluminum, nickel, titanium, and 13 other alloys and has accumulated a large amount of data. After the 1980s, Chengdu Hongbo Industrial Co., Ltd., and Jinduicheng Molybdenum Co., Ltd., also introduced a Φ250 mm three-roll Y-type rolling mill for tungsten and molybdenum wire production from KOCKS [146]. This changed the domestic backward tungsten and molybdenum wire processing technology, product quality, and performance in the poor situation. In 1998, Luoyang Nonferrous Metals Processing Design and Research Institute developed a Φ370 mm three-roll Y-type rolling mill and successfully used it for rolling large-size sintered tungsten and molybdenum bars. In 1983, the University of Science and Technology in Beijing prepared for the construction of a three-roll cold-rolling continuous rolling unit steel wire production line, followed by the development of the second generation of continuous rolling units, and the Ministry of Metallurgy technical appraisal completed product inspection [147].

The three-roll Y-type mill hole system, according to the different uses, is divided into two types: the extension hole type and the finishing hole type [148]. The compression of each pass of the extended hole system is large, and the rolled piece is close to the shape and size of the finished product after a number of passes. The fine-rolled hole type system has a small compression amount in each pass and regularizes the rolled parts rolled through the extended hole type to obtain the final product with the required shape and size. According to the geometry of the hole type, the extended hole type is categorized into three types: arc–triangle hole type, flat–triangle hole type, and arc–triangle–circle hole type.

The arc–triangle hole type is known for its uniform stress and deformation, reduced roll wear, and suitability for less plastic metals. Its pass elongation is high; the first pass is recommended to be 1.15~1.25, and the subsequent ones can have unified elongation. However, the curvature of the arc close to the round increases free torsion, affecting stability [149]. The flat triangle hole type is characterized by a significant reduction in roll wear, roll torsion, and good stability, but the stress concentration is prone to sharp corners and not applicable to low-plasticity metals. Its elongation coefficient range is wider, up to 1.25~1.60, and it is suitable for round tungsten–molybdenum bar initial rolling [150]. The arc–triangle–circular hole type system combines the characteristics of stability and excellent shape of the finished product, but the elongation coefficient is relatively small: 1.20~1.25. In practice, attention needs to be paid to the tension balance between the racks, and there are limitations on the generality of the metal with widely varying width and spreading properties [151]. The fine-rolling hole system is further refined to include the finished hole and its pre-processing hole, with a round hole mainly supplemented by the arc–triangle hole to realize high-precision processing. The system is configured into four categories to meet different processing needs. Feng Yaya et al. [139] proposed the “flat triangle-arc triangle-circle” hole type and “alternating positive and negative Y-shape” rack arrangement. It combines the advantages of stability, low wear, and direct molding, and it is especially suitable for online coiling of tungsten–molybdenum rods, which is achieved by staggering the racks by 180° to achieve torsion-free rolling.

Non-ferrous tungsten needs to be rolled at high temperatures, which can lead to splitting of the rolled parts if the material is not plastic enough. Compared with the curved triangular hole type, the flat triangular hole type ensures a uniform internal organization of the rolled parts during the rolling process and provides excellent spreading performance at high temperatures. The three-roll-type system is unique in that there is no fixed main groove, and each type operates as if it were a main groove. As shown in Figure 1, for a flat or nearly flat three-roll hole type, the inner circle diameter is used as the defining reference for hole calculation [152].

The inlet and outlet areas of the three-roll frame set are always circular in design, and the initial machining stage is often characterized by the use of a flat triangular hole type, which creates an asymmetric hexagonal cross-section in the first pass. Subsequently, a carefully designed oval and pre-rounded three-roll hole type is gradually shaped into a finished circular groove with two or three different radii, ensuring both shape and precision. As shown in Figure 2, at least three passes are required to form the final circular cross-section [152].

In the hot rolling of tungsten alloys, the Y-type flat triangular hole type system can be used throughout the bar-rolling process. The three rolls are distributed at 120°, and by flexibly adjusting the angle of the rolling position of each roll in the plane, even under the change of the flat hole type with different internal diameters, the hexagonal cross-section can be ensured to be rounded gradually to realize an efficient and accurate rolling process, as shown in Figure 3 [152].

Zhong Guisong and Liu Jiarong et al. [153] of Beijing Nonferrous Metals Research Institute (BNMRI) studied the hole type system for the continuous rolling of tungsten–molybdenum strips in a three-roll Y-type mill. To achieve good plasticity of molybdenum tungsten at high temperatures, they selected the “flat triangle—round hole type” system and provided key calculation formulas for hole type calculation, rolling area, and rolling tension. Lv Yanjun et al. [154] studied the hole design method of thirteen passes of “arc-triangle-circle hole type” continuous Al bar rolling, introduced the adjustment of the hole parameters and the hole adjustment method, and carried out experimental verification. Kang Jin et al. [155] optimized the design of a hole type for fifteen passes of three-roll continuous rolling of aluminum rods. A “flat triangle-circle” hole system was adopted to replace the “curved triangle-circle”. By reasonably setting the extension and filling coefficients and calibrating the tension coefficient, the mechanical properties of the rolled material were significantly improved, and the production cost was reduced. In summary, the bar blanks need to go through a number of passes through the hole-rolling deformation, and the cross-section shape and size and performance of the key to achieve accuracy of the hole design is accurate and optimized. The reasonableness of the hole design is directly related to the quality of the finished product, mill productivity, product cost control, improvement of labor conditions, and reduction in labor intensity. In the design of billet mill roll holes, the first indispensable premise is to deeply understand and accurately grasp the technical specifications of the product, the basic parameters of the billet, the detailed technical parameters of the mill, the motor’s load-bearing capacity, and the performance characteristics of the supporting equipment. Subsequently, the need to carefully select the appropriate hole system, such as billet open billet box holes, is often used to ensure that the rolling process is smooth and efficient. Next, the design of a reasonable pressure for the program is critical. It determines the precise size and shape of the exit cross-section of the rolled parts in each rolling pass, which is the key to realizing the specification requirements of the final product. Finally, based on the results of the above design and elaborate drawings of the hole type and rolls, these drawings are not only the direct basis for production operations, but also an important technical document to ensure stable product quality and smooth production processes. Through this series of scientific and rigorous design process, we can maximize the rolling efficiency, reduce production costs, and produce high-quality bar products in line with high standards.

### 3.3. Roll Service Environment and Damage

Zhao xin, Li wenping et al. [156] pointed out that in cold rolling of tungsten alloy plates, the deformation resistance is far higher than in hot rolling. Rolls bite into the surface by more than 10,000 MPa of pressure and are accompanied by shear, high temperatures, shock loads, and rolling defects. Therefore, cold rolls need to have at least 2000 MPa tensile strength, and the internal organization needs to be pure and uniform. Sun Guifang et al. [157] pointed out that when hot rolling bars, the rolls are subject to cyclic thermal stress, contact stress, shear stress, and residual stress, but also, they are subject to high-temperature rolling heat radiation and oxide layer wear. Therefore, the hot mill rolls need to have high strength, high hardness, high temperature resistance, impact resistance, wear resistance, and resistance to thermal cracking, among other characteristics. The two failure forms of tungsten alloy hot-rolling rolls are spalling of the roll surface caused by various factors such as thermal cyclic stress, tensile stress, and plastic strain and wear of the roll surface caused by thermal fatigue and repeated friction. Zang Qingfeng et al. [158] concluded that the traditional surface repair technology makes it difficult to enhance the high-temperature wear and crack resistance of the rolls, while the laser surface treatment has a significant effect, forming nanocrystalline zones to enhance the strength and the bonding force. In the hot-rolling process of sintered tungsten billets, in order to extend the life of the rolls and reduce the problems of thermal cracking, oxidation, and fracture of the roll surface, effective measures need to be taken to protect the rolls. Li Ying [159] pointed out that the roll surface cooling is divided into internal and external technology. Internal cooling through the roll core peripheral design is key to reducing the roll surface temperature; its internal structure is also key. External cooling using spray coolant, curved piping, and the inert gas cooling effect is good, and the form of the coolant piping and the media type are key to improving the efficiency.

Hot rolling takes place under high pressure and high temperatures, and these extreme conditions lead to stress concentrations that accelerate roll wear and crack expansion. Roll surface degradation directly affects the surface quality of the rolled material, increasing scrap rates and the need for rework. Weidlich F [160] elaborated that the roll surface is restricted by the thermal expansion and contraction of the material to produce thermal stresses, and the residual stresses left over from the manufacturing of rolls and the mechanical stresses in the molding process work together to affect the performance and life of the rolls comprehensively. In hot rolling, the thermal fatigue effect is more significant than the mechanical stress fatigue, and the mechanical stress is significantly affected by factors such as bar cross-section shape, undercut rate, and rolling temperature. Fan X. B. [161] describes how mill vibration can lead to product quality degradation, equipment failure, and damage. Although structural defects in the mill are the main cause of vibration and are difficult to suppress directly, the adjustment of rolling parameters is more operable. Among them, the rolling force increases with the pressure ratio and rotational speed and decreases with the temperature, and the effect of the pressure ratio is more significant. Therefore, the production should be given priority to optimize the allocation of under-pressure to control vibration. Yan X. Q [162] pointed out that the existing vibration suppression means are mostly passive adjustments of mechanical, hydraulic, electrical, and rolling parameters. In view of the complex multi-parameter, multi-field coupling, and non-linear characteristics of rolling mill vibration, it is difficult for the mathematical model to accurately reflect the actual dynamics, which leads to ineffectiveness of some of the suppression methods. To this end, an innovative active vibration suppression strategy is proposed, which utilizes a perturbation estimation and compensation algorithm to treat the vibration-triggering forces and model errors as total perturbations and accurately estimates the compensable perturbations in the control inputs of the servo valves through an extended state observer in order to achieve efficient vibration suppression.

The durability of rolls, as key tools in the plastic deformation of metals, is the primary criterion for the efficiency of metal-forming technology. This criterion is particularly important for metals such as tungsten and molybdenum, which are refractory and highly resistant to deformation. S. M. Gorbatyuk et al. [163] and his team detailed an innovative technology from the Moscow Institute of Steel and Alloys that focuses on the process of manufacturing semi-finished products from rotary rolled tungsten–molybdenum billets. It was found that wear occurred during the initial rolling of the billets using standard 50-steel rolls with a carbon content of 0.5%. Subsequently, 40 Kh steel rolls with an increased hardness of 45–50 HRC were used to improve durability; however, even at this hardness, the rolls showed signs of wear after processing only 25–30 kg of tungsten billets. In response to the challenges of high thermal and mechanical loads in the rolling process, the international community is committed to the development of carbide rolls suitable for high-temperature rolling of difficult-to-deform materials. The CR-95 carbide introduced by Sandvik Coromant of Sweden, containing 70% WC and 30% Co, is effective against thermal cracking, and its life expectancy is 18 times that of traditional steel rolls, with tool consumption cut in half. Germany’s Hertel, on the other hand, uses special alloy work rolls based on WC with Co or Ni, which show excellent resistance to thermal cracking. In Russia, highly durable rolls are mostly made of tungsten–cobalt carbide VK-8 and VK-20, which have become the material of choice for rolling tungsten alloys of hard-to-deform metals by virtue of their ultimate bending strength of 1700 MPa, high hardness of 70–85 HRC, and lack of welding at high temperatures.

## 4. Effect of Process Parameters on Continuous Rolling

In the continuous rolling process, the four key parameters that affect product quality are the pass press amount, rolling temperature, roll speed, and rolling method. The setting of the amount of compression needs to take into account the nature of the material, process requirements, equipment capacity, and safety. Usually, the amount of compression bypass decreases to ensure uniform deformation and quality of the finished product. In cold rolling, the plastic deformation of the metal is directly affected by the amount of compression, and a press amount that is too high or too low can damage the material, such as by causing fractures or cracks. In addition, the rolling force and temperature affect the amount of compression. Rolling force is determined by the radial and axial forces of the rolls and the strength of the material; temperature affects the hardness and plasticity of the material, indirectly regulating the amount of undercut. In order to ensure high efficiency and quality of rolling, there is a need for real-time monitoring and adjustment of the amount of pressure. At the same time, the size of the rolled product, surface quality, and mechanical properties of the comprehensive testing and analysis are important in order to quickly find and solve the problem, along with achieving a timely adjustment of the rolling parameters.

### 4.1. Effect of Pass Press Amount on Continuous Rolling

Due to the narrow plastic temperature range and large deformation resistance of the tungsten and its alloys, its total processing rate needs to be rolled and then combined with several anneals to be completed. To prevent oxidation and improve efficiency, the total processing rate between anneals should be maximized within the capacity of the mill. The processing rate of the passes needs to be moderate. If it is too small, it is prone to uneven deformation and delamination cracking, especially since the initial rolling passes should be larger. With the increase in the total deformation rate, the rolling temperature decreases and resistance increases. Therefore, the processing rate should be decreasing; hot rolling should use 20–30% of the amount of compression and warm rolling should use 10–20% of the amount of compression. Cold rolling involves significant work hardening, large deformation, and ease of cracking, so the pass processing rate should be less than 10% [164].

W plate properties are significantly affected by the rolling process, and control of the undercut is especially critical [165]. In addition, the rolling press amount can also affect the microstructure of the material, such as the texture of the sum of different grain orientations, and certain properties of tungsten are obtained through the texture evolution, thus controlling its mechanical properties. The typical technique for forming the desired texture in metals is usually a combination of annealing processes between plastic deformation and mechanical deformation steps [166,167]. Zhang et al. [135] performed unidirectional rolling of pure W powders with different thicknesses (18 mm to 3 mm) after cold isostatic pressing, pre-sintering, and densification by mid-frequency sintering, and they were annealed in hydrogen at 137 K for 2 h. The sintered W powders were then annealed at 137 K for 2 h in hydrogen. It was shown that the sintered tungsten was initially randomly oriented. During the hot rolling process, the tungsten grains underwent fragmentation, nucleation growth, and fibrillation. The texture varied with different rolling pressures: 40%, 60%, and 90% pressures formed more γ-fiber textures, while 80% showed θ-fiber and Goss textures. The microhardness, flexural strength, and relative density increased with increasing rolling pressure; the 60% pressure rate samples had the highest thermal conductivity and the best impact properties, while the 80% pressure rate may have exhibited the best irradiation resistance. Zhao et al. [61] prepared W-0.5 wt%ZrC (WZC) specimens with 46%, 69%, 77%, and 96% press amounts, respectively, using a multi-step rolling technique. The effects of different press amounts on the properties of WZC materials were evaluated by Vickers hardness and tensile tests, as shown in Table 1. Figure 4 illustrates the RD-ND planar metallographs, showing that the W grains were significantly elongated in the RD direction. In the WZC14 sample, the average size of the RD and ND grains was 26.2 μm versus 12.5 μm, with an L/D ratio of about 2.1. With the increase in the rolling press rate from 69% to 77%, the grains were further deformed, with a decrease in the size of the ND and an increase in the size of the RD. In the 96% rolled WZC01 sample, it was difficult to distinguish grain boundaries due to severe elongation, the hardness increased with the increase in the press amounts, and the ND grain size and the ductility-free temperature decreased with the increase in the press rate.

Li et al. [168] prepared pure and doped tungsten sheets with a thickness of 36.5 mm using a powder metallurgy method by heating them at temperatures above 1450 °C and subsequently rolling them to 26.6 mm, 19.7 mm, 10.9 mm, and 3.6 mm by unidirectional rolling at rolling pressure rates of 27%, 46%, 70%, and 90%, respectively. The aim was to elucidate how the rolling depression and impurity elements affect the microstructure evolution. The results show that both pure tungsten and doped W undergo grain crushing, recrystallization, grain growth, and fibrillation. In contrast, pure tungsten at 46% shows a bimodal structure with fine and large grains. In doped tungsten, the dragging effect of the impurities on dislocations or grain boundaries effectively suppresses the abnormal grain growth during the dynamic recrystallization process, thus promoting a more moderate recrystallization process. Both 70% rolled pure W and 90% rolled doped W exhibit a fibrous structure, with dominant textures <111>//ND and <100>//ND. The bimodal recrystallization profile is the key to the formation of fibrous crystals, while the uniform recrystallization profile of doped W interrupts the continuity of the fiber structure and exacerbates the formation of transverse cracks. Figure 5 shows the microstructural evolution of the hot-rolling process as the rolling depression proceeds, including crushing, restitution, recrystallization (nucleation and grain growth), slip, and fibrillation. Initially, sintered isometric crystals are crushed under anisotropic stress to form dislocation substructures and provide the driving force for hot rolling. Secondly, with the increase in press amount and temperature, sub-crystals of dislocation rearrangement are formed during the restitution process, and the energy stored in the form of dislocations and small-angle grain boundaries within the deformed organization promotes the nucleation process of continuous dynamic recrystallization. Finally, under high temperature and high rolling pressure, some special grains produced by some recrystallization in the early stage grow abnormally to form abnormal recrystallization and finally form a bimodal structure where coarse abnormal grains and fine dynamic recrystallized grains exist at the same time. Under the strong interaction, the slip system inside the coarse grains is activated, and the distribution of hard massive grains between the coarse grains triggers stress inhomogeneity, which in turn dominates the slip behavior. Under this effect, the soft, coarse grains are twisted by the hindrance of the hard, fine grains, and eventually, a fibrous structure is formed.

In summary, the key effect of rolling pressure is significantly reflected in the regulation of the microstructure of the alloy material, including grain morphology, size, texture, and dispersion distribution. This control mechanism in turn optimizes the mechanical properties of the material, such as Vickers microhardness, tensile strength and ductility, etc., which highlights the importance of optimizing the rolling process parameters to enhance the material properties.

### 4.2. Effect of Rolling Temperature on Continuous Rolling

Tungsten and its alloys have high melting points and are difficult-to-deform metals, which are generally plasticized by hot rolling. The rolling temperature is divided into the opening rolling temperature and the final rolling temperature, and according to the processing passes, the corresponding passes are selected for temperature drops. The initial rolling temperature directly affects the deformation of the rolling process. The final rolling temperature, stress–strain, and rolling force, especially the effect of residual stress and residual strain, determine the mechanical properties of the rolled parts after rolling. In addition, an initial rolling temperature that is too high not only does not improve the performance of the rolled parts, but also causes energy waste [169]. The hot-rolling process covers both hardening and softening phases, and work hardening can be effectively mitigated by recrystallization during the softening phase. However, hardening may lead to incomplete recrystallization, coexisting recrystallization, and state organization processing. Recrystallization in dynamic equilibrium promotes the formation of equiaxial crystals, enhances tungsten processing toughness, and significantly reduces residual stresses and dislocation density. Reddy et al. [170] analyzed the atomistic simulation of the nano-rolling process of nanocrystalline tungsten and showed that increasing the rolling temperature decreases the total dislocation density and increases the atomic fraction of the twinned grain boundaries. During low-temperature rolling, localized shear bands nucleate from the surface or grain boundaries and propagate along the grains to accommodate plastic strains due to rolling deformation. Rolling at higher temperatures leads to dynamic recrystallization, which produces larger equiaxed grains, and hot rolling also contributes to the formation of sub-grain boundaries with slight orientation changes. Wang J et al. [171] carried out a thermal simulation of polycrystalline tungsten at 1250~1550 °C and 0.001~1 s^−1^ and constructed its processing map. At low temperatures and high strain rates, the uneven local deformation leads to stress concentration and triggers thermal processing instability. At high temperatures and low strain rates, dynamic recovery and recrystallization compete with grain growth, leading to uneven fiber organization and deformation instability. Domazet Željko et al. [172] pointed out that rolling speed is the core parameter in hot rolling, which directly affects yield, processing time, productivity, and material properties. Rolling speed needs to be adjusted at each stage. In the opening stage, low speed and high pressure are required to ensure smooth biting and condition change. The intermediate stage is used to prevent the final rolling temperature from being too low and to protect efficiency by using high-speed rolling. In the final stage, medium speed is used to balance heat transfer and size control to avoid excessive temperature differences or shape distortion.

### 4.3. Effect of Rolling Method on Continuous Rolling

Strain path changes significantly affect material deformation and texture. A single rolling direction leads to excessive grain elongation, strengthens the rolling direction but weakens the vertical direction, and generates anisotropy, leading to transverse cracking. Switching rolling directions can induce bi-directional deformation of grains from fibrous to planar pancakes, which are interwoven vertically and horizontally, reducing anisotropy, improving deformation uniformity, and eliminating deformation inhomogeneity defects [173,174]. Zhang et al. [165] used three rolling methods: UNR (unidirectional rolling), CRR (cross rolling), and CLR (clock rolling). As shown in Figure 6, after about 80% thinning over 8 passes, the products were annealed in hydrogen at 1100 °C for 2 h. It was found that UNR, CRR, and CLR all formed θ and γ fiber textures, but UNR exhibited a moderate recrystallization level, a significant θ fiber texture, and a weaker γ fiber texture. The high texture level and crack tip resulted in UNR’s lower bending strength. In order to obtain a low-crystallization-index texture and high strength, the moderately deformed UNR is preferred to CRR and CLR.

Xiong et al. [175] used Ni-W-Co-Ta MHA bars to produce plates with uniform microstructures via orthogonal rolling through the LG-300 mill. This method significantly improves the organization uniformity and reduces the texture and property anisotropy compared with conventional unidirectional and cross-rolling. The orthogonal rolling plate strength increased by 100 MPa, with a tensile breaking elongation of up to 13.8%, far more than the unidirectional rolling of 8.7%.

In summary, the strain path of a material during processing has a significant effect on its deformation behavior, texture formation, and final properties. A single rolling direction (such as UNR) tends to lead to excessive grain elongation and significant anisotropy, which in turn reduces the transverse properties of the material and increases the risk of cracking. In contrast, by switching the rolling direction (such as CRR or CLR) or adopting orthogonal rolling, the bidirectional deformation of the grains can be effectively induced so that the morphology of the grains is transformed from fibrous to longitudinally and horizontally interwoven planar pancakes, which significantly reduces the anisotropy and improves the deformation homogeneity and the overall properties of the material. By precisely regulating the rolling path, rolling speed, rolling temperature, and other parameters, the texture distribution and performance of the material can be further optimized to meet the demand for high-performance materials in different fields.

### 4.4. Evolution Mechanism of Heat Treatment on the Microstructure of Deformed Metals

Metamorphic metals are in a sub-stable state due to stored energy, and the free energy is elevated. During annealing, the atomic diffusion capacity is increased, which leads to the transition to a stable state and consequent changes in properties. The annealing process is prolonged with the temperature rise (beyond the critical temperature of tungsten alloys) and time length and undergoes three stages of recovery, recrystallization, and grain growth [176].

During the annealing process, the vacancies and dislocations move and adjust their number and configuration. At low temperatures, vacancies migrate and disappear at surfaces, grain boundaries, or dislocations, reducing the density of vacancies and significantly lowering the resistivity with few changes in mechanical properties. At high temperatures, the dislocations are rearranged and annihilated by slip, climb, and cross-slip to form more stable configurations [177]. Hetero-sign dislocations annihilate and homo-sign dislocations rearrange to form low-energy grain boundaries or polygonization, which reduces the storage energy. Tungsten material forms a cytosolic structure after multi-slip deformation, the vacancies are reduced in the early stage of annealing, the intracellular dislocations slip toward the wall to offset the hetero-sign dislocations, and then the cell wall dislocations form a low-energy lattice, which evolves into a sub-crystalline structure. Further annealing promotes the coarsening of sub-crystalline grains and reduces the interfacial energy, which is the process of grain growth driven by the reduction in the interfacial area.

Recrystallization is the process of generating and growing new distortion-free grains in a deformed metal, driven by a decrease in stored energy. The process is divided into two phases: nucleation and growth, with polygonization as a preparation for nucleation. Nucleation occurs at high-angle grain boundaries, preferentially in high-storage-energy regions such as grain boundaries, trident nodes, and deformation-induced zones. The new nucleus interface advances towards the deformation zone until it stops meeting neighboring grains. The growth rate of the nuclei is affected by both driving force and resistance, with fast growth at high initial storage energy and then slowing down as the storage energy decreases. Eventually, recrystallization is complete when the grains meet [178].

After complete recrystallization, the grains grow by mutual annexation, allowing the crystal to reach a lower free energy state. The reduction in the grain boundary area reduces the internal free energy, and the driving force for grain growth originates from the interfacial energy difference, especially the stored energy at high-angle grain boundaries. In normal growth, fine grains tend to become larger due to high interfacial energy. However, boundary-pinning effects (such as small particles and low-mobility, low-angle grain boundaries) can lead to anomalous grain growth [179].

## 5. Texture Evolution and Property Analysis of Tungsten and Its Alloys after Hot Continuous Rolling and Annealing

Tungsten and its alloys are used in a wide range of applications, including lighting engineering, the electronics industry, manufacturing, aerospace, military, the medical industry, and nuclear energy. The demand for their properties varies from field to field, and these properties are strongly and significantly affected by the crystal texture [180]. Specifically, key indicators such as irradiation resistance [181,182,183], surface modification under particle bombardment [184,185], fracture toughness [130], brittle transition temperature, and even the energy reflection coefficient of particles [186] are highly dependent on the specific orientation of the crystals. Therefore, precise modulation of the crystal texture of tungsten alloys is essential for optimizing their properties to meet the demands of diverse applications. Recent research has shown that tungsten grains with specific crystal orientations exhibit excellent resistance to irradiation when bombarded with energetic particles such as tritium, helium, and heavy ions. This discovery provides new ideas and possibilities for improving the stability and durability of materials in extreme radiation environments. For example, the surface morphology after irradiation correlates with the normal direction of the particles [184]. The densely rowed <111> direction exhibits the greatest thermal conductivity [187]. The {110} cracked system exhibits higher fracture toughness at room temperature compared to the {100} cracked system [135]. The texture consists of the distribution of grain orientations and tungsten properties can be optimized by modulating the evolution of texture. Commonly used techniques combine plastic deformation and annealing with thermo-mechanical treatments to guide the lattice rearrangement in preferred directions [167]. Unidirectional rolling is used as a simple method to promote tungsten texture transformation. For effective monatomic tungsten, it is certain that the addition of additives increases the elongation to failure and enhances the toughness of the system. However, the real questions that need to be investigated is the “how” and “why” of system control. The question is why the basic physical mechanisms introduced by the additive are so effective in toughening the alloy. Therefore, in order to examine these mechanisms, it is reasonable to introduce characterization techniques in order to gain a comprehensive understanding of the microstructure and its response to applied stimuli. EBSD is commonly used to determine single or multiple particle orientations, whereas X-ray diffractometry generalizes the overall picture of particle orientation over a large area. Therefore, EBSD and X-ray diffraction are more suitable for determining micro-textures and macrotextures, respectively [188]. In the coordinate system of cubic crystal rolled samples, (hkl) [uvw] is often used to express the orientation of a grain. The metallic polycrystalline organization with a selective orientation produced after deformation is the deformation texture. The texture that appears after recrystallization and annealing is the recrystallization texture. The texture that appears after recrystallization and annealing is the recrystallization texture. Rolled tungsten is a typical bcc rolled texture consisting of a ϒ-weave (parallel to the {111}<uvw> grain plane) and an α-weave (parallel to the {hkl}<110> grain orientation) [189]. The main types of textures in BCC metals are described below [190].

(1)*α-Fiber* (crystallographic fiber axis <110>parallel to the rolling direction, including major components: {001}<110>;{112}<110>;{111}<110>).(2)*ε-Fiber* (crystallographic fiber axis<011> parallel to the transversal direction, including major components: {001}<011>;{112}<111>;{111}<112>; {011}<100>).(3)*ϒ-Fiber* (crystallographic fiber axis<111> parallel to the normal direction, including major components: {110}<110>;{111}<112>).(4)*η-Fiber* (crystallographic fiber axis<100> parallel to the rolling direction, including major components: {001}<100>;{011}<100>).(5)*θ-Fiber* (crystallographic fiber axis<001> parallel to the normal direction, including major components: {001}<100>;{001}<110>).(6)*ξ-Fiber* (crystallographic fiber axis<011> parallel to the normal direction, including major components: {011}<100>;{011}<211>;{011}<111>; {011}<011>).

In the hot-rolling process of tungsten alloys, the rolling method, rolling deformation, diffuse phase, and recrystallization annealing are important factors that affect the type of texture component of the material as well as the strength of each [180]. Since the crystal texture is produced by thermomechanical processing and largely determines the anisotropy of the material properties, understanding the development of the texture is essential for designing end products with enhanced properties [135,191]. For the texture of rolled materials, the microstructure of plates and bars is analyzed by scanning electron microscopy (SEM) (grain size, sub-grain, crystal structure, etc.) and transmission electron microscopy (TEM) (bright field imaging, scanning TEM, etc.). For its characterization, most of the polar figure (PF), inverse polar figure (IPF), grain boundaries (GBs) figure, local average misorientation (LAM) figure, and orientation distribution function (ODF) are used to analyze the microstructural evolution. The IPF maps crystal orientations differentiated by color, the GB figure shows the crystal size and morphology, and the LAM figure reveals the orientational correlations between adjacent grains. Li et al. [168] analyzed the microstructural evolution of hot-rolled pure and doped tungsten by means of an IPF map, GB map, and LAM map, as shown in Figure 7. The sintered state was homogeneous polycrystalline with random grain orientation. After 27% rolling, sub-structures were present within the processed grains with blurred grain boundaries. By 46%, the small-angle grain boundaries decreased, signaling the beginning of recrystallization and the formation of large-angle grain boundaries. Pure tungsten recrystallized rapidly under high strain and high temperature, and the grain grew significantly, showing a bimodal structure, while doped tungsten was delayed to 70% due to the stabilizing effect of impurities. With 90% rolling, both of them were due to the high strain refinement of grains, increased grain boundaries, and formation of a fibrous structure.

The microstructure and texture evolution of tungsten alloys recrystallized under high-temperature annealing is accurately described in terms of grain size, aspect ratio, and volume fraction of fiber weave composition as quantitative indicators. Wang et al. [192] investigated the microstructure and texture evolution of tungsten plates containing 2 vol% yttrium oxide hot-rolled to a 50% thickness reduction during annealing between 1200 °C and 1350 °C, as shown in Figure 8 and Figure 9. Polar diagrams are a crystallographic method of studying materials by projecting their crystallographic orientations and their interrelationships onto an imaginary sphere and studying them against a standard polar diagram. The polar diagram in Figure 8 shows the polar diagrams of partially and fully recrystallized samples after annealing at 1250 °C. The diagrams show that the texture changed gradually from concentrated to fuzzy to diffuse during the annealing process, the intensity of the texture changed from strong to weak, and the texture began to be fuzzy at 48 h, which indicates that recrystallization occurred at this time in comparison with the antipodal diagrams and the performance of the microstructure diagrams. Figure 9 shows the orientation distribution function for performing quantitative texture analysis. The cell grain orientations parallel to the rolling direction and the grain faces parallel to the rolling surfaces are known from the antipodal and polar plots. In order to obtain the determination of the crystallographic orientation relationship of the grain faces, the data were further processed to obtain the ODF map and compare it with the aggregated regions of the metal grain orientation in space, and information about the relevant grain faces could be obtained. Figure 9d reveals that the warm-rolled sample exhibited a typical rolling texture of body-centered cubic metal containing α-fibers (<110> along the RD), γ-fibers (<111> along the ND), and weaker θ-fibers (<001> along the ND). The average length-to-diameter ratio of the grains remained stable during recrystallization, and the recrystallized texture exhibited a higher degree of randomness compared to the deformed texture.

Due to friction and residual stresses during the rolling of the plates, the development of the crystal texture may vary from the surface to the whole so that there is severe full-thickness texture inhomogeneity. Kumar et al. [193] thoroughly investigated the microstructure and crystal texture changes in a heavy alloy of pure tungsten embedded in a nickel–iron tungsten matrix after cold rolling. It was found that the macroscopic shear bands originate from the microscopic shear bands inside the crystals at the initial stage of deformation, and their directions are affected by the initial orientation of the crystals. During cold rolling, the tungsten phase shows weak α- and γ-fiber textures at low rolling, but the α-fibers are significantly enhanced with deeper rolling. The matrix phase forms weak β-fibers with low rolling, connecting multiple textures. Goss- and Brass-oriented fibers are significantly enhanced with high rolling. These findings have important implications for optimizing the cold-rolling process and alloy properties. The DBTT of W alloys is mainly affected by crystal defects, crystal orientation, impurity status, and preparation route [13,64,194]. The initial recrystallization temperature (RCT) mainly depends on the grain structure and foreign impurity (elemental/second phase particles) status [195]. In addition, composition and fabrication routes have significant effects on both DBTT and RCT [196]. Zhao et al. [197] prepared WYZ alloy plates by hot rolling, in which Zr and Y_2_O_3_ synergistically formed a stable Y-Zr-O phase, which enhanced the interfacial strength and improved the alloy strength and toughness. After rolling, the alloy exhibited different texture characteristics at each crystal plane, such as weak α and Goss fibers at the RD-TD plane; strong Goss and weak θ fibers at the RD-ND plane; and strong θ, α, and γ fibers at the TD-ND plane, which led to anisotropy in properties. Low-deformation rolling with Y-Zr-O doping rose the initial recrystallization temperature of the alloy to 1400 °C, and the texture strength was enhanced after annealing. In particular, the alloys exhibited excellent tensile strength and ductility due to the TD-ND planar dislocation advantage in the RD orientation.

Although the evolution of the texture of thin plates in unidirectional and cross-rolling is well studied, there need to be more studies on shape rolling, especially for bar-rolling processes. Bar rolling, due to its unique sample deformation state, may significantly alter the strain paths, which in turn affects the choice of operating slip system, an area that needs to be explored in depth [198,199]. The design of the bar-rolling process directly affects the distribution of the final microstructure, and in order to achieve the desired properties, the grain size needs to be finely regulated from two dimensions, grain refinement and homogenization, to optimize the overall properties of the material [200]. Grain refinement follows the Hall–Petch relationship, and the number of grain boundaries increases dramatically after refinement, which effectively disperses dislocation sources and reduces stress concentration, thus synergistically enhancing the plasticity and strength of the material. In particular, fine-grained tungsten exhibits excellent irradiation resistance, and its large grain size can effectively absorb irradiation-induced point defects to ensure the stability of material properties. Large-sized TiC particles are uniformly embedded in the grain boundaries, which are combined with plastic deformation to refine the grains, significantly improving the strength and toughness of tungsten materials. The recrystallization temperature of pure tungsten is low, and the grain is easy to enlarge under high rolling. The addition of TiC particles improves the recrystallization temperature so that the W-TiC alloy maintains fine crystals under high rolling and optimizes the performance [201]. Bar rolling, which is sub-dynamic recrystallization-dominant, needs to regulate the cooling rate and rolling temperature to refine the grain. For homogenization, local grain size inhomogeneity is the performance bottleneck, and the key process parameters cover temperature, strain, strain rate, and initial tissue state. Shao et al. [199] investigated the effect of multi-pass hot rolling on the microstructure of doped tungsten and found that the hardness was enhanced by grain refinement, but there was weird grain growth near the center. This originated from the strain energy release of the super potassium bubble pinning force, resulting in grain boundary migration. Adjusting the rolling temperature and rolling speed reduces the stress unevenness and optimizes the organization’s uniformity. Non-uniform strain distribution leads to high strain concentration, which affects the longitudinal intergranular strength [202]. Therefore, it is necessary to pay attention to the localization of grain boundary stresses and second-order residual stresses.

## 6. Summary

Tungsten and its alloys are widely used in several fields, and their properties are significantly affected by the crystal texture. For example, irradiation resistance, fracture toughness, thermal conductivity, etc., are all closely related to the crystal orientation, so specific properties of tungsten alloys can be obtained by controlling the texture evolution. Hot rolling is a key step in the manufacturing process of tungsten alloys. However, in actual production, due to the combined effect of a variety of factors, such as temperature, elastic deformation of rolling mill components, plastic deformation of materials, mill vibration, etc., it is difficult to control the grain size, surface quality, and product dimensional accuracy effectively, which leads to a decline in mechanical properties; therefore, optimizing the rolling process parameters to improve rolling stability and productivity has become an urgent problem. In the pre-rolling manufacturing process, the development of powder production and solidification technology provides the possibility of obtaining ultra-fine and nanocrystalline structures. Among them, mechanical alloying and discharge plasma sintering are the key technologies. In addition, the addition of elemental particles to the powder provides a key basis for enhancing alloy properties. During hot rolling and annealing, crystal reversion and recrystallization can significantly affect material properties. After complete recrystallization, grains grow by merging and merging, leading to a decrease in internal free energy, but abnormal grain growth occurs in the presence of boundary pegging effects such as small grains or low-angle grain boundaries. Microscopic characterization techniques such as SEM and TEM can analyze the microstructural evolution of plates and bars, including grain size, sub-grains, and crystal structure. At the same time, a combination of analytical methods such as polar diagrams, inverse polar diagrams, grain boundary diagrams, and orientation distribution functions can provide a comprehensive understanding of the microstructure and its response to external stimuli. In summary, this paper systematically analyzes the effect of microstructure evolution on the properties of tungsten and its alloys during the molding process from multiple perspectives, which provides an important reference for the further optimization of tungsten alloys’ manufacturing process and properties. Future research should not only focus on the relationship between the microstructure and properties of tungsten alloys and how to optimize the properties through fine control, but also further explore new alloy design and preparation methods with an aim to obtain more excellent tungsten-based materials. The current process flow of tungsten alloy rolling forming is summarized in Table 2.

## Figures and Tables

**Figure 1 materials-17-04531-f001:**
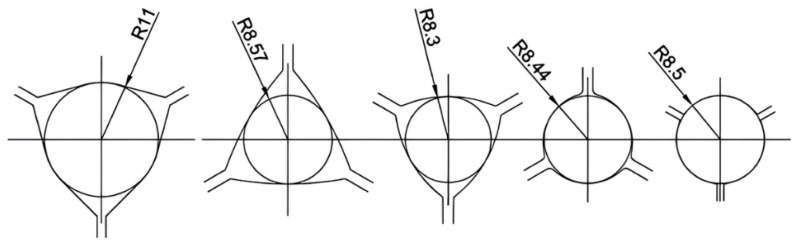
Hole types with different inner circle diameters [152].

**Figure 2 materials-17-04531-f002:**
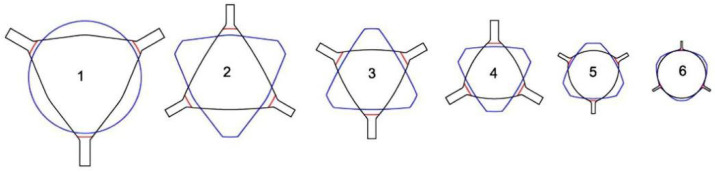
Three-roll rolling pass hole system for bar diameters from 28 mm to 16 mm; 1–6 show the rolling diagram of bar rotating 120 degrees with each other [152].

**Figure 3 materials-17-04531-f003:**
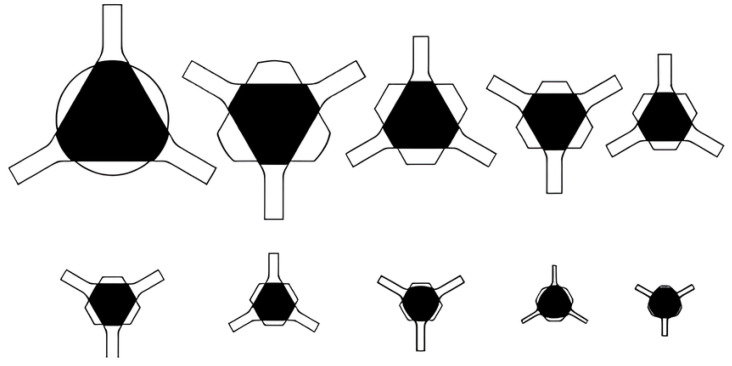
Three-roll flat hole type bar-rounding system [152].

**Figure 4 materials-17-04531-f004:**
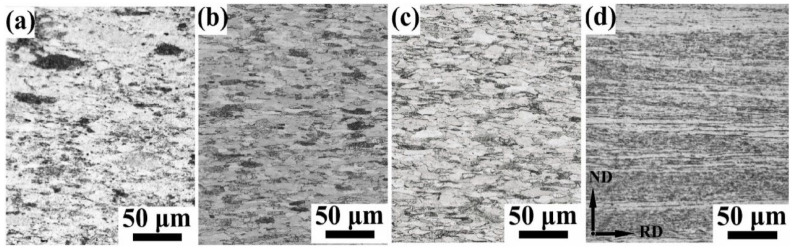
Metallographic images of RD-ND planes of (**a**) WZC14, (**b**) WZC08, (**c**) WZC06, and (**d**) WZC01 [61].

**Figure 5 materials-17-04531-f005:**
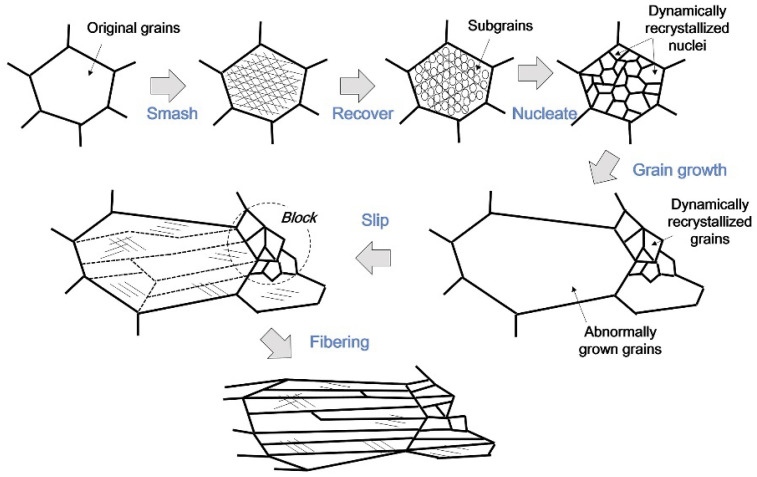
Diagram of microstructure evolution of tungsten during hot rolling [168].

**Figure 6 materials-17-04531-f006:**
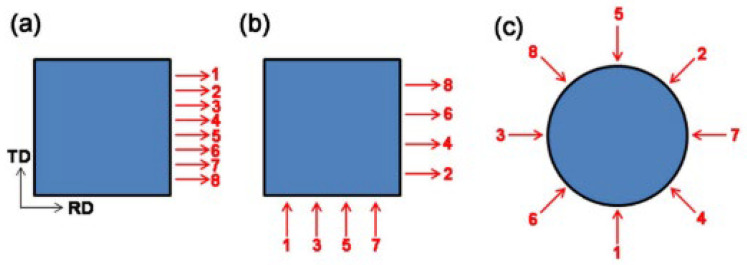
Schematic diagrams of UNR (**a**), CRR (**b**) and CLR (**c**); 1–8 for the same rolling direction in UNR; 1–8 for the rolling direction of clockwise rotated 90° and 270° alternately in CRR; 1–8 for the rolling direction of clockwise rotated 135° in CLR [165].

**Figure 7 materials-17-04531-f007:**
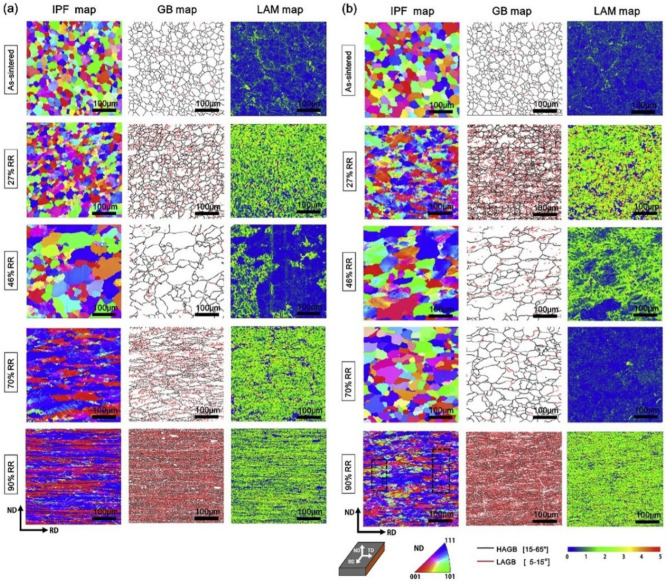
EBSD maps of the hot-rolled pure W (**a**) and doped W (**b**) under various rolling reductions (RR = 0%, 27%, 46%, 70%, 90%) in transverse direction (TD) views, presented as an inverse pole figure (IPF) map, a grain boundaries (GB) map, and a local average misorientation (LAM) map [168].

**Figure 8 materials-17-04531-f008:**
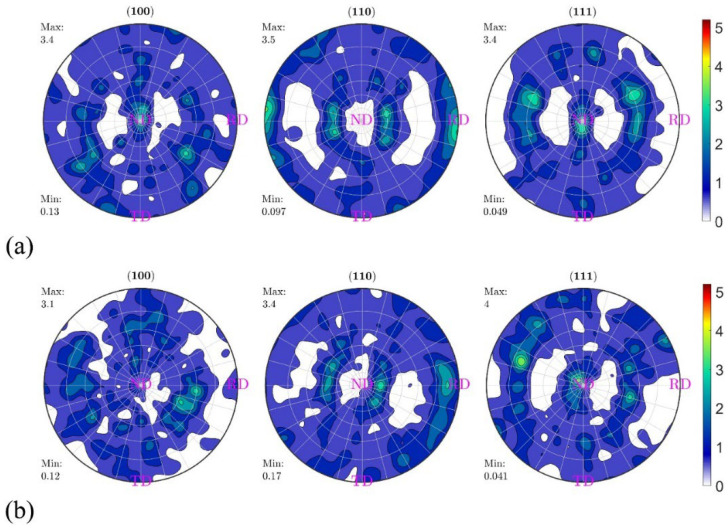
Pole figures of warm-rolled tungsten with 2 vol% Y_2_O_3_ (WY50 plate) after annealing at 1250 °C: (**a**) partially recrystallized sample annealed for 48 h and (**b**) fully recrystallized sample after 84 h [192].

**Figure 9 materials-17-04531-f009:**
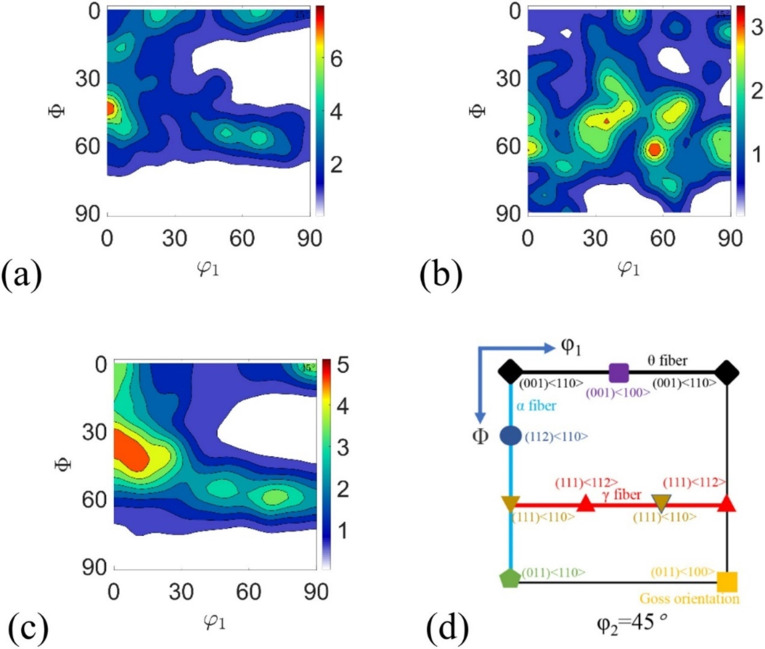
Quantitative texture analysis of tungsten with 2 vol% Y_2_O_3_ for different conditions illustrated by the φ_2_ = 45° section of orientation space: orientation distribution functions of partially recrystallized samples annealed at 1250 °C for (**a**) 48 h and (**b**) 84 h, and (**c**) the warm-rolled sample; (**d**) schematic illustration of important texture components in bcc metals [192].

**Table 1 materials-17-04531-t001:** Grain size, Vickers hardness, and NDT value of rolled WZC materials with different rolling reductions [61].

Samples	Rolling Reduction	Average Grain Size (ND, μm)	Average Grain Size (RD, μm)	Aspect Ration	Vicke, Hardness (Hv)	NDT (°C)
WZC01	96%	0.48	-	-	508.9 ± 10.6	100
WZC06	77%	3.89	14.4	3.7	486.1 ± 9.7	150
WZC8	69%	4.26	13.6	3.2	466.9 ± 9.7	150
WZC14	46%	12.5	26.2	2.1	440.2 ± 8.9	200

**Table 2 materials-17-04531-t002:** Tungsten alloy rolling forming process.

	Macro-Level							Micro-Level		
	Pre-Rolling Process				Rolling Process			Recovery	Recrystallization	Grain Growth
Tungsten and Its Alloys	process	Additives	Micro-powder manufacturing	Powder densification	Rolling Mill	Two-roll rolling mill	Three-roll rolling mill	Dislocation annihilation and sub-grain boundary formation	Deformation-free nucleus formation and nucleus growth	High-angle grain boundary formation and abnormal grain growth
	technology	Alloying elements, rare earth oxides, ceramic oxides, carbides, etc.	mechanical mixing method, chemical method, etc.	SPS, UHP, HIP, HP, and PLS, etc.	Hole Type	Square, diamond, oval, and circle, etc.	Arc triangle, flat triangle, and arc triangle-circle, etc.			
					Process parameters	Pass press amount, rolling temperature, rolling method, etc.				
	theory	Particle strengthening and Fine grain strengthening			Deformation strengthening, dispersion strengthening, grain boundary strengthening			Heat treatment strengthening		
	Performance	Improves processing plasticity, toughness, and material hardness			Improves material strength, plasticity, and toughness			Hardness and strength increase. plasticity and toughness decrease		
	Characterization techniques	OM, SEM, TEM, EDX, EBSD, PF, IPF, LAM, and ODF, etc.								

## Data Availability

The original contributions presented in the study are included in the article, further inquiries can be directed to the corresponding author.

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
