# Peer review of "Research Progress on Rolling Forming of Tungsten Alloy"

_materials, 2024, doi:10.3390/ma17184531_

Round 1

Reviewer 1 Report

Comments and Suggestions for Authors

This article provides a review on the research progress of tungsten alloy rolling formation. The article is in pretty decent shape although there are some important aspects that need to be resolved.

1. There is some odd text that reads, "Error! No text of specified style in document.."

2. For figures, need to make sure that you get proper copyright.

3. Summary section is severely lacking content. This section needs to be greatly improved such as providing a summary table.

Reviewer 2 Report

Comments and Suggestions for Authors

The information contained in the article extends knowledge about the processes of manufacturing and improving the properties of high-temperature alloys. It was rightly noted that: (i) improving the properties of tungsten requires an interdisciplinary approach, combining material engineering with advanced characterization techniques; (ii) microstructure and texture control is key to achieving the desired mechanical and physical properties for tungsten alloys; and (iii) production stability and efficiency can be increased through accurate modeling and production process optimization.

Author Response

Comments 1: [improving the properties of tungsten requires an interdisciplinary approach, combining material engineering with advanced characterization techniques;] 

Response 1:  Thank you for pointing this out. I agree with this comment. Advanced characterization techniques (e.g. SEM, XPS, etc.) are used to characterize and analyze the microstructure and surface properties of tungsten materials, so that it can be used to guide the optimization of tungsten material preparation process and performance improvement. By combining materials engineering and advanced characterization techniques through an interdisciplinary approach, comprehensive improvement and precise control of tungsten material properties can be achieved.

Comments 2: [microstructure and texture control is key to achieving the desired mechanical and physical properties for tungsten alloys;]

Response 2: Thank you for pointing this out. I agree with this comment. Fine control of microstructure and texture is essential to achieve the excellent mechanical and physical properties required for tungsten alloys. By adjusting micro features such as grain size, shape, orientation, and phase distribution, the strength, toughness, wear resistance, and thermal stability of tungsten alloys can be significantly enhanced to meet the stringent requirements of high-end applications.

Comments 3: [production stability and efficiency can be increased through accurate modeling and production process optimization.

Response 3: Thank you for pointing this out. I agree with this comment. Precise modeling can accurately predict material properties and guide alloy design. Production process optimization can reduce scrap rate, improve production efficiency and material utilization, and ensure stable and reliable product quality. The combination of the two is the key to efficient and stable production of tungsten alloys.

Reviewer 3 Report

Comments and Suggestions for Authors

 At the beginning of page 9 there is no reference source. 

At the end of page 10 a similar error with respect to Figure 3-1.

Analogous editorial errors can be found on page 20.

Few spelling errors (e.g., on page 12).

Reviewer 4 Report

Comments and Suggestions for Authors

The manuscript describes recent achievements and progresses in rolling of tungsten and tungsten alloys. The work is quite interesting but it is very long and of not easy readability. This is the main drawback of the paper and this aspect must be improved before publication.  For instance, the introduction (four pages) is too long; in more concise way the authors should describe the main problems and possible approaches to solve them. Moreover, a flow-chart of the treated matter could be of help.

 The authors should also pay attention to the used terminology; in the paper there are many cases where right concepts are expressed not rigorously. For instance:

- PFM is the acronym of plasma facing material, not plasma-oriented material, as written in the introduction (8th row).

- “Brittleness is common in body-centered cubic cells, mainly because there are few independent slip systems in body-centered cubic cells[16]”. BCC structure is more correct because plastic deformation involves the whole lattice not single cells.

Etc..

Finally, I suggest to have a careful revision of the text to improve English.

Comments on the Quality of English Language

I suggest to have a careful revision of the text to improve English.

Round 2

Reviewer 1 Report

Comments and Suggestions for Authors

The authors have done a decent job to address the comments, however, the error message was still found in the manuscript so please correct it. Also, why were roman numerals used instead of typical numbers for the reference numbers?
